# Funding and policy incentives to encourage implementation of point-of-care C-reactive protein testing for lower respiratory tract infection in NHS primary care: a mixed-methods evaluation

Matthew Johnson,[1] Liz Cross,[2] Nick Sandison,[3] Jamie Stevenson,[3] Thomas Monks,[1] Michael Moore[4]

¹NIHR CLAHRC Wessex Data Science Hub, Faculty of Health Sciences, University of Southampton, Southampton, UK
²Attenborough Surgery, Bushey Medical Centre, Herts Valleys Clinical Commissioning Group, NIHR CLARHC East of England, Bushey, UK
³NIHR CLAHRC Wessex, Faculty of Health Sciences, University of Southampton, Southampton, UK
⁴Primary Care and Population Sciences, Faculty of Medicine, University of Southampton, Southampton, UK

**Correspondence to**
Matthew Johnson;
mj1c13@soton.ac.uk

## ABSTRACT

**Objectives** Utilisation of point-of-care C-reactive protein testing for lower respiratory tract infection has been limited in UK primary care, with costs and funding suggested as important barriers. We aimed to use existing National Health Service funding and policy mechanisms to alleviate these barriers and engage with clinicians and healthcare commissioners to encourage implementation.

**Design** A mixed-methods study design was adopted, including a qualitative survey to identify clinicians' and commissioners' perceived benefits, barriers and enablers post-implementation, and quantitative analysis of results from a real-world implementation study.

**Interventions** We developed a funding specification to underpin local reimbursement of general practices for test delivery based on an item of service payment. We also created training and administrative materials to facilitate implementation by reducing organisational burden. The implementation study provided intervention sites with a testing device and supplies, training and practical assistance.

**Results** Despite engagement with several groups, implementation and uptake of our funding specification were limited. Survey respondents confirmed costs and funding as important barriers in addition to physical and operational constraints and cited training and the value of a local champion as enablers.

**Conclusions** Although survey respondents highlighted the clinical benefits, funding remains a barrier to implementation in UK primary care and appears not to be alleviated by the existing financial incentives available to commissioners. The potential to meet incentive targets using lower cost methods, a lack of policy consistency or competing financial pressures and commissioning programmes may be important determinants of local priorities. An implementation champion could help to catalyse support and overcome operational barriers at the local level, but widespread implementation is likely to require national policy change. Successful implementation may reproduce antibiotic prescribing reductions observed in research studies.

## Strengths and limitations of this study

► Use of a mixed-methods study design to assess the benefits, barriers and enablers of implementation from multiple perspectives.
► The study did not involve research funding for participating sites to enable evaluation of the impact of real-world financial structures associated with National Health Service (NHS) commissioning.
► Development of a pack of resources that could contribute to future implementation projects.
► The study was undertaken against a background of general financial constraint within the NHS, which may have adversely impacted on outcomes.

## BACKGROUND

Acute uncomplicated lower respiratory tract infection (LRTI) is the one of the most common acute illnesses managed in primary care, and even in low antibiotic prescribing countries most patients receive antibiotics.[1 2] There is a clear national and international agenda to reduce unnecessary antibiotic prescribing.[3] The recently updated Cochrane review[4] of antibiotics for acute bronchitis demonstrated modest benefits, with a reduction of cough duration of around half a day. These findings were not replicated in a recently published large trial of antibiotics against placebo.[5] Limited benefit was demonstrated from antibiotics likely to be balanced by harms, and no subgroup was identified in whom there was a clinically relevant benefit.[5 6]

In the absence of clear benefit then what are the drivers of continued prescribing? Patients are concerned about their symptoms,[7] and clinicians are worried about missing severe infection and to avoid medicolegal consequences.[7–9] However, continued prescribing

of antibiotics carries direct prescribing costs, increased reconsultations[10] and the major threat of antibiotic resistance.[11] Moreover, a large cohort study has shown that adverse events following primary care consultation with patients with LRTI are rare and may not be directly influenced by prescribing strategy.[12]

There is evidence that antibiotic prescribing in LRTI may be limited by appropriate use of near-patient tests (NPTs).[13–15] Two candidates are available: C-reactive protein (CRP) and procalcitonin (PCT).[16–18] An individual patient data review and meta-analysis supported the use of PCT to guide antibiotic use in acute settings including primary care, emergency units and intensive care, and demonstrated equivalent clinical outcomes with reduced antibiotic uptake.[19] Similarly, a recent Cochrane review examining the role of CRP in acute respiratory illness in primary care[20] included six trials with 3,284 participants and demonstrated a reduction in antibiotic use, although the results were interpreted with caution due to a high degree of heterogeneity. The recently published National Institute for Clinical Excellence (NICE) pneumonia guidelines[21] have also endorsed the use of CRP to aid decision making in primary care, selecting this ahead of PCT given the current non-availability of an NPT for PCT.

Several trials have explored the use of CRP in the primary care setting for management of LRTI, either alone or in combination with a communications skills training package, and have demonstrated a substantial reduction in antibiotic prescribing.[13–15] Although CRP is widely used in Scandinavian countries, uptake has been limited in the UK, despite evidence of effectiveness in trial contexts to direct rational prescribing for LRTI. There is some question, however, of the effectiveness of CRP once adopted in clinical practice; results of tests performed on those with upper respiratory tract infection were found to have been misinterpreted, and modest effects on prescribing described.[22] Some have questioned whether reduced antibiotic prescribing will be seen following implementation in low prescribing settings,[23] while others have reported CRP being the main determinant of antibiotic prescription in observational cohorts.[24]

The reasons for the delayed uptake of NPT in the UK are not clear. Tests to reduce diagnostic uncertainty were supported by primary care physicians in a multicountry study including the UK.[25] Although studies suggest that CRP is a cost-effective means of addressing LRTI in primary care, there is evidence that concerns around costs and funding remain a barrier to widespread implementation.[26] As the UK National Health Service (NHS) model of primary care does not include item of service payments, implementation of point-of-care (PoC) testing outside of a research setting would generate additional work and costs for initial purchase, maintenance and consumables, whereas antibiotic prescriptions have no direct cost at practice level (see supplementary appendix A for further detail of primary care testing in the NHS system). One plausible way to increase utilisation of CRP PoC would be the introduction of an item of service payment for use of the test in management of LRTI. The NHS England General Medical Services contract, in addition to defining the scope of standard primary care services to be delivered by general practices, also includes provision for opt-in to the delivery of additional, 'locally enhanced' services (LES).[27] This study was based on the hypothesis that the LES scheme may provide a mechanism to introduce a financial incentive to uptake of CRP PoC for the management of LRTI in an NHS primary care setting.

We aimed to evaluate the efficacy of an item of service payment framework introduced at the local level by way of the LES scheme as a means of encouraging implementation among clinicians and healthcare commissioners. We also aimed to work with other groups and localities to explore alternative approaches to implementation and to identify the perceived benefits, barriers and enablers using a post-implementation survey.

## METHODS
### Leveraging funding and policy incentives
Our work has concentrated on making use of the opportunities afforded by existing NHS funding and policy mechanisms to encourage implementation of CRP PoC in primary care. We did not provide any research funding to participating organisations to ensure that successful implementation was not artificial, and could potentially be reproduced by others in the context of the real-world financial structures and constraints associated with healthcare commissioning in the NHS. All work in this area was undertaken during 2015 and 2016.

We developed a standard LES specification to underpin local implementation, establishing a funding framework of reimbursement of general practices by Clinical Commissioning Groups (CCGs) for CRP PoC on a unit basis. In view of the importance given to budgetary concerns by commissioners considering CRP implementation,[26] CCGs may be motivated by its potential to open access to national funding associated with achieving the NHS England 'Quality Premium' (QP) target for reduced antibiotic prescribing in primary care.[28]

Our research group, National Institute for Health Research (NIHR) Collaboration for Leadership in Applied Health Research and Care (CLAHRC) Wessex, is funded by both the NIHR and partner organisations (including CCGs) within the local health system. Partner funding contributions may be monetary, or comprised of research study involvement. Our locality covers nine CCGs, each of whom had the opportunity to fulfil this funding obligation by participating in a CRP implementation study, or similar research. As well as this benefit, there was further opportunity for any participation costs to be partially or fully offset if the QP was achieved as a result.

### Engaging with the NHS

Using materials from the GRACE Intro study,[15] we developed resources including an online training course for general practitioners on the use of CRP, a clinical audit form and patient information leaflet.[29] All resources were made available to interested organisations as a means of facilitating implementation by reducing the associated administrative burden.

We visited clinicians and healthcare commissioners in our locality to generate interest and gave presentations at locality events to promote the LES framework. We also attended an NIHR CLARHC Wessex showcase event to which local CCGs were invited. We followed up additional enquiries from other groups outside of our locality who were interested in CRP implementation by offering visits and presentations and sharing the resources developed for our local study. Resources were shared with ten groups across the country.

### Post-implementation survey

In August 2017, following our period of NHS engagement, we issued an electronic survey to a convenience sample of clinicians and commissioners who had expressed an interest in, or were known to have contributed to, CRP implementation projects. Overall, 19 individuals were invited to participate, including healthcare commissioners, pharmacists, primary and secondary care clinicians and public health professionals.

We adopted a qualitative approach to explore in more depth the factors motivating respondents' initial interest, their experience of the implementation process and perceived barriers and enablers. Survey questions were written in line with these underlying objectives as deductively generated main themes[30] (box 1).

Following the method of thematic analysis described by Nowell and colleagues,[31] three members of the research team (MJ, NS and TM) individually reviewed all survey responses to inductively identify more specific subthemes. Reviewers took a systematic and iterative approach to analysis, later using researcher triangulation to reach consensus.

---

**Box 1  Post-implementation survey questions**

**We asked participants:**
► What were your/your organisation's reasons for implementing C-reactive protein (CRP) testing?
► What was your experience of implementing and using CRP testing, and what is happening now?
► Which aspects of the implementation worked well?
► What were the barriers to implementation and/or continued use?
► How did you overcome these barriers?
► What would have helped, or would help in the future to encourage continued use?
► What would facilitate the implementation process?
► What would be your recommendations for those looking to implement CRP testing in the future?

---

**Box 2  Herts Valleys CCG implementation study**

Funded by an National Health Service (NHS) England Innovation Challenge Prize, an implementation study was undertaken in Herts Valleys CCG to evaluate C-reactive protein (CRP) utilisation over three winter months (November 2016–January 2017) and in five general practices, purposively sampled using standardised practice-level prescribing data to target high and medium antibiotic prescribers. The study aimed to evaluate whether, compared with standard care, the availability of CRP PoC for LRTI in primary care was associated with reduced acute and follow-up antibiotic prescribing and unscheduled primary care reattendances and healthcare contacts in the 28 days following presentation.

Participating practices received an intervention consisting of one testing device and supplies to perform 100 tests, training on the National Institute for Clinical Excellence (NICE) guidelines and equipment use, a review visit and practical assistance from the study team where appropriate; all other costs were borne by the practice. Each practice was free to select an appropriate device location and means of operationalising patient flow based on the physical layout of the practice, available resources and staff skill mix.

In line with the NICE guidelines, patients aged 18-65 years presenting to intervention practices with suspected lower respiratory tract infection (LRTI) of less than 3 weeks' duration where there was diagnostic uncertainty were eligible to receive a test. Eligibility was assessed by the clinician during patient consultation. Patients with acute pneumonia, pregnant, immunocompromised, terminally ill or under follow up for chronic obstructive pulmonary disease were excluded.[21] As the offer was made on clinician discretion, and the patient entitled to refuse, some eligible patients did not receive a test. However, all eligible patients presenting to intervention practices were included in the evaluation, irrespective of whether they received a test.

The five intervention practices were compared with three Herts Valleys CCG control practices of similar size and prescribing level, all of which continued to provide standard care. Control practices did not receive training. One member of the study team (LC) conducted a retrospective electronic search at control practices to identify new clinical consultations (during the same study period) with patients who met the CRP eligibility criteria. Presentations were identified using a set of Read codes[38] commonly used to record clinical activity related to LRTI in NHS primary care, and relevant information collected for analytical purposes. Results from the implementation study are given in supplementary appendix B.

---

### Implementation case study

In parallel with our work to evaluate the use of an item of service payment framework as a means of encouraging CRP implementation, a separate study was undertaken in Herts Valleys CCG to evaluate CRP utilisation over three winter months (November 2016–January 2017). This case study did not use the LES framework, being separately funded by an NHS England Innovation Challenge Prize and driven by a local champion. However, in view of the successful implementation in this locality, we present further detail in box 2 and results in supplementary appendix B to demonstrate the potential effects of implementation of CRP PoC on antibiotic prescribing.

### Patient and public involvement

There was no patient and public involvement (PPI) in development of the research question, although

implementation of CRP PoC flowed from the NICE pneumonia guidelines,[21] the development of which involved substantial PPI input. There was no PPI in development of the LES specification. This would not be normal practice in respect of a contractual arrangement for the funding of general practices.

## RESULTS

### Adoption of the LES framework and implementation of CRP

While there was initial interest in CRP PoC facilitated by use of the LES framework, ultimately no CCGs within the NIHR CLAHRC Wessex locality participated in implementation projects. CRP was under consideration by one local CCG as part of a range of measures that might contribute to achieving a 'Quality, Innovation, Productivity and Prevention' programme target around improving detection of pneumonia in primary care, with the aim of enabling earlier intervention and reducing hospital admissions. The CCG had planned to implement CRP across all of its general practices, but concluded that the associated upfront capital cost was too substantial and did not proceed.

Another CCG outside of our locality was interested in more widespread CRP implementation based on antibiotic prescribing reductions observed during a pilot undertaken in a single general practice. Although 10 testing devices were procured and were initially regularly used, declining utilisation in the face of operational barriers prompted the CCG to cease procurement of PoC consumables. Financial incentivisation by way of the LES framework was considered as a means of encouraging utilisation but ultimately failed to re-engage interest.

We are not aware of any other CCGs having adopted the LES framework or having engaged in implementation projects.

### Post-implementation survey

Of the 19 individuals invited to participate, 7 (37%) submitted full responses. Several subthemes emerged from inductive analysis, with a high level of consistency among respondents (box 3).

All respondents reported being organisationally motivated by the potential for CRP PoC to help reduce antibiotic prescribing, while some further specified a desire to reduce variation in prescribing rates among practices in their locality. However, respondents also described mixed clinician utilisation: while some regularly incorporated PoC into consultations for suspected LRTI, others did not use it at all. Furthermore, one respondent noted that while utilisation had initially been high, it had declined over time.

### Benefits

Most respondents agreed that CRP is a valuable clinical aid to appropriate antibiotic prescribing for patients with symptoms of LRTI. Furthermore, some highlighted its value as an objective measure to improve patient

---

### Box 3    Benefits, barriers and enablers of implementation

**Benefits:**
- ► A clinical aid to appropriate antibiotic prescribing.
- ► An objective measure to improve patient confidence in the prescribing action.

**Barriers:**
- ► Limited time available during consultation.
- ► Layout of facility and placement of testing device.
- ► Cost of implementation and continued use.
- ► Source of funding.
- ► Resistance to change.
- ► Maintaining engagement.

**Enablers:**
- ► Early adopters to share experience and provide mentorship.
- ► Training and education.
- ► Champions within practice/locality.
- ► Collaboration at local and national level.
- ► Better utilisation of IT to facilitate testing process.

---

confidence in the chosen prescribing action, particularly in consultation with those who are 'very keen' to receive antibiotics. Two respondents noted that, in their experience, patients had responded positively to the test and were satisfied with the outcome.

### Barriers

In general, respondents reported that interest among clinicians was sometimes poor and suggested a need for financial incentives and support to encourage widespread uptake. Most mentioned cost pressures, while some questioned who should be responsible for funding: general practices or the CCG. Despite the evidence base for the clinical benefits, one respondent suggested that there remains a need to 'clearly demonstrate short term benefits in costs, workload and safety' to develop and maintain engagement.

Most respondents commented on the impact of operational constraints, such as the physical layout of the practice, how to accommodate multiple users and the time required to carry out the test, particularly in the context of high workload and limited consultation duration. Although some respondents argued that other benefits justified its use despite these barriers, others specifically cited them as disincentives, especially for clinicians who may have a negative attitude to CRP or be resistant to change.

### Enablers

Most respondents discussed the importance of collaboration, although interpretations of this differed. Some suggested that early adopter sites share lessons learnt to help others and avoid duplicated effort. The value of training and education during the implementation process were consistently emphasised, and development of a standard programme was suggested. Others mentioned the role of NIHR in fostering collaborative

working and the potential for general practice or CCG champions to improve engagement and resolve problems. Some respondents also suggested better use of IT to facilitate testing. Specific examples included the deployment of standard templates to record the test and result in the practice management system and the use of electronic alerts during consultation to prompt clinicians to PoC if indicated.

## DISCUSSION
### Summary of main findings
Despite initial interest, there was no implementation in the NIHR CLAHRC Wessex locality, and no CCGs formally adopted the LES framework. The research team were unable to gain significant traction with CCG management, and when contact was established CCGs were unwilling to prioritise antibiotic stewardship over other local initiatives. The policy levers seemed to have little impact in this locality, where CCGs were struggling to remain in budget. The financial rewards arising from the QP only applied to CCGs meeting financial targets. Elsewhere, one CCG implemented CRP and, following declining utilisation in response to operational barriers, found that the LES framework was insufficient as a mechanism to re-engage interest.

Although the small sample size limits inference and generalisability, our post-implementation survey identified several financial, operational and physical barriers in common with previous qualitative research.[26] Respondents confirmed that implementation would be unlikely without financial incentives but also highlighted difficulties integrating PoC into practice workflow, and constraints arising from a lack of dedicated space, equipment sharing and limited time. Reported enablers included adequate training and the value of a local champion.

Some respondents also emphasised the clinical benefits of CRP, giving anecdotal examples of cases where testing had prevented antibiotic prescription. The potential for more widespread repetition of this outcome is suggested by quantitative results from the Herts Valley CCG implementation study, where a successful, separately funded implementation scheme was run for a 3 month period, driven by a local champion. Observation of substantial prescribing reductions among intervention practices suggests that implementation in the NHS might replicate the prescribing reductions reported in research studies.

### Comparison with other literature
We are unaware of any other implementation studies concerning CRP PoC in the UK. In other health settings, PoC is widely adopted,[22] and following government directives has been introduced in the Netherlands.[32] The financial barriers to implementation have been identified in a previous study including European and UK participants,[23] which noted that countries with high rates of use had alternative reimbursement models and that widespread implementation in Europe followed health policy

change. The same study also highlighted issues around workflow and time as potential barriers to implementation in the UK.

### Strengths and weaknesses
Our study describes the results of attempts at CRP implementation without the resources associated with research and without specific policy directives. It is unclear how generalisable our findings might be; it would appear that CCG partnership with NIHR CLAHRC Wessex and national-level incentives via the QP should have maximised the potential for local implementation. The scheme was devised during a time of general financial constraint within the NHS, which may have had particular impact in the Wessex locality.

### Limitations around funding mechanisms
The criteria required to achieve the QP, even taking the antibiotic prescribing element alone, has been inconsistent. Some changes have been significant, such as a move to greater emphasis on antibiotic prescribing for urinary tract infection as of 2018/2019.[28] Furthermore, as the QP is awarded retrospectively and is contingent on meeting other financial targets, the funding mechanism is not guaranteed, making it difficult to engage commissioners and to create a firm financial framework to underpin CRP implementation.

A further feature of the QP is that no method of achievement is stipulated; the antibiotic prescribing element simply requires an absolute prescribing reduction. The NHS has reported a national ~7% reduction in primary care coinciding with the implementation phase of this study,[33–35] which may have resulted from a general policy shift and increased focus of clinical training in primary care. This suggests that overall improvements could be gained and the QP target potentially achieved by way of alternative, lower cost methods alone, negating commissioners' financial incentive for CRP implementation irrespective of the clinical benefits.

The pressures of multiple, competing commissioning programmes may limit engagement with certain initiatives, while the overall funding structure of the NHS may also influence commissioners' preferences and priorities. One CCG within our locality suggested that, despite evidence of a net cost saving associated with CRP,[36] while the upfront implementation costs reside with primary care, any savings would principally be realised by the secondary care sector. In this instance, therefore, concerns that the costs and benefits of the initiative may be distinctly localised within separate areas of the health system acted as a disincentive to its adoption.

### Implications
While the use of existing financial structures appeared appealing as a mechanism, it was not possible to fully test the hypothesis that modest financial incentives to general practices at local level would enable CRP implementation, as financial pressures impeded CCG adoption of

the policy. National incentives for CCGs did not appear to override the financial constraints because: (A) financial rewards were only available to CCGs meeting financial targets and (B) antibiotic targets were being achieved through other mechanisms not requiring financial investment.

Although a small case study suggests that implementation outside of research studies may result in similar prescribing reductions, since it was driven by local investment and a local champion, it may not fully reflect implementation in routine practice or be generalisable to other areas. Furthermore, and recalling questions over the primacy of lower cost measures, the fact that this intervention provided training and support in addition to testing materials limits the extent to which the observed prescribing reductions can be confidently attributed to CRP PoC alone.

The value of an enthusiastic, local champion to catalyse support for implementation emerged from both the qualitative and quantitative strands of this study. Knowledge mobilisation and implementation in practice may be assisted by way of a Researcher-In-Residence model,[37] while further qualitative and observational research could improve understanding of how champions are able to persuade and engage clinicians and to encourage commissioners to look beyond the immediate financial disincentives, and whether they may be effective in other areas and settings. Further economic research might also model different modes of implementation to assess the costs and consequences across the system and to find alternative funding models to overcome the financial barriers. Multipurpose testing devices, for example, may have the advantage of spreading investment across several funding streams.

In conclusion, it seems unlikely that financial schemes falling outside of national policy will gain much traction in a financially constrained NHS. Full-scale implementation of CRP PoC is likely to require central implementation via government policy or contractual changes.

**Acknowledgements** The authors would like to thank Elsie Griffins and Dr Denise Knight for their support for the Herts Valleys CCG implementation study.

**Contributors** MJ carried out quantitative analysis and wrote the paper with contributions from MM. MJ and NS developed the postimplementation survey and, with TM, carried out qualitative analysis. LC carried out the Herts Valleys CCG implementation study. JS, NS and MM developed the LES framework and other resources and engaged with the National Health Service. TM provided methodological input, and MM provided clinical guidance. All authors commented on drafts of the paper and have read and approved the final manuscript.

**Funding** This research was funded by the National Institute for Health Research (NIHR) Collaboration for Leadership in Applied Health Research and Care Wessex at University Hospital Southampton NHS Foundation Trust. The Herts Valleys CCG implementation study was separately supported by a 2015/16 NHS England Innovation Challenge Prize, Acorn award.

**Disclaimer** The views expressed are those of the author(s) and not necessarily those of the NHS, the NIHR or the Department of Health and Social Care.

**Competing interests** MJ, NS, JS, TM and MM have no competing interests to declare. LC has received honoraria from Abbott Laboratories and Roche Diagnostics Ltd for speaking events.

**Patient consent** Not required.

**Ethics approval** The Integrated Research Application System confirmed that formal ethical approval was not required for the Herts Valleys CCG implementation study, which is a service evaluation project.

**Provenance and peer review** Not commissioned; externally peer reviewed.

**Data sharing statement** No additional data available.

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
