## [Reviewer comments · BMJ Open]

ARTICLE DETAILS

TITLE (PROVISIONAL)	Funding and policy incentives to encourage implementation of point-of-care C-reactive protein testing for lower respiratory tract infection in NHS primary care: a mixed-methods evaluation
AUTHORS	Johnson, Matthew; Cross, Liz; Sandison, Nick; Stevenson, Jamie; Monks, T; Moore, Michael

VERSION 1 – REVIEW

REVIEWER	John P. Hays Associate Professor, Erasmus University Medical Centre Rotterdam (Erasmus MC), the Netherlands
REVIEW RETURNED	08-Jun-2018

GENERAL COMMENTS	An interesting attempt by Johnson et al. to encourage the implementation of CRP PoC testing for LRTI in NHS primary care in the UK. Please comment on the following points: 1) 'Most respondents agreed that CRP is a valuable clinical aid to appropriate antibiotic prescribing for 203 patients with symptoms of LRTI.' Before the study, did the authors record the views of the participating respondents with respect to the applicability of CRP for diagnosing infections (CRP specificity for infection as opposed to CRP being perceived as a general biomarker for inflammation)? This would allow respondent stratification and comparison of 'open-minded' versus 'sceptical' mindsets to be researched (at least for future research and if more interest is shown in the CRP PoC concept). If not, do the authors think that these opinions could have affected the participation in, and results of, the study?2) In the Abstract: Results : 'The implementation study demonstrated reduced rates of antibiotic prescription and follow-up consultation amongst intervention sites.' These results are 'hidden' in Appendix A. I wonder if this should be part of the main body text?3) Can the authors say anything about the infection burden during the study period? For example, was there (excess) influenza circulating in the communities sampled? Would the use of CRP be viewed as more valuable during an influenza outbreak and could this possibly affect the views of respondents (in this study)?4) I wonder if the authors could possibly comment on, or have considered, the implementation of influenza and/or respiratory syncytial virus PoC testing instead of CRP PoC testing for LRTI in primary care? Would this be more interesting than CRP?5) Respondents understand about the association between non-targeted antibiotic prescribing and increased antibiotic resistance, but this may seem as an abstract concept without solid evidence about the (cost) effectiveness of non-targeted antibiotic prescribing
---

	within their own practice. Have the authors considered utilising information on the incidence of antibiotic resistance within the local community and then specifically focusing their NPT implementation efforts to regions where the (threat of) antibiotic resistance is highest? 6) I think that the 'use of electronic alerts during consultation to prompt clinicians to PoC if indicated.' Is an interesting suggestion. However, unless there is a standard consensus on what criteria actually warrant the use of a PoC (not forgetting cost calculations, test specificity for a particular disease, the actual 'weight' of the PoC result in the individual doctor's decision-making process etc), the development of such algorithms may not be as easy to implement as initially appears. 7) The idea of utilizing an 'enthusiastic, local champion to catalyse support for implementation' (in this case for CRP PoC testing) is an interesting idea, but who will the champion represent – Respondents, financial authorities, pharmacists, PoC diagnostic developers or patients? These different potential users may have conflicting views. 8) Within the framework of a 'local champion', it may be worth remembering that local hospitals may represent an untapped pool of knowledge and tips on PoC implementation, having possibly gone through the process of assessing the implementation of PoC themselves (https://www.futuremedicine.com/doi/pdf/10.2217/fmb-2016-0120). 9) Reference 29 does not lead to the 'online training course for general practitioners on the use of CRP, a clinical audit form and patient information leaflet' mentioned in the text. This material should be made available for reviewers and readers.
--	--

REVIEWER	Hasse Melbye General Practice Research Unit in Tromsø, UIT the Arctic University of Norway
REVIEW RETURNED	22-Jun-2018

GENERAL COMMENTS	This is a study on an attempt to implement CRP test in British general practice. Why the attempt did not succeed was examined by a qualitative study with a strategic sample of interviewees. The study should be of particular interest for GPs in UK, and of primary care policy makers in UK and other countries. For most readers outside Britain the study may appear less relevant. The background for the study is well described, but it would be of help to get more key information on British primary care, for instance how the performance of laboratory tests is organized. In other countries, at least in Scandinavia, an office laboratory is running all the day, and laboratory tests are something GPs refer to, not carry out themselves. The manuscript is generally clearly written, but knowledge on NHS and the funding of primary care in UK would help the reader. Main concern. The study carried out in Herts, is presented in Box 2 and the appendix. It is not described in other parts of the manuscript, and its role in a potential paper remains too unclear. The "no" for statistics in the review check list is due to the missing description of methods for Herts study. Other comments. In the qualitative part of the study, time constraints and facility and placement of testing device were listed among the barriers. In the discussion, the funding attained most of the focus. As a reviewer from outside Britain it seems strange that the organization of laboratory services in British primary care escapes attention.
---

VERSION 1 – AUTHOR RESPONSE

Reviewer: 1 Reviewer Name: John P. Hays An interesting attempt by Johnson et al. to encourage the implementation of CRP PoC testing for LRTI in NHS primary care in the UK. Please comment on the following points: 1) 'Most respondents agreed that CRP is a valuable clinical aid to appropriate antibiotic prescribing for patients with symptoms of LRTI.' Before the study, did the authors record the views of the participating respondents with respect to the applicability of CRP for diagnosing infections (CRP specificity for infection as opposed to CRP being perceived as a general biomarker for inflammation)? This would allow respondent stratification and comparison of 'openminded' versus 'sceptical' mindsets to be researched (at least for future research and if more interest is shown in the CRP PoC concept). If not, do the authors think that these opinions could have affected the participation in, and results of, the study?	No we did not collect views prior to the survey. The sample was drawn from clinicians who had expressed an interest in the test and from commissioners so perhaps it is not surprising that respondents were generally in favour of the test itself. The barriers identified by respondents were organisational rather than objections to the test itself. The purpose of the questionnaire was to identify barriers and enablers to implementation. We would have sought a different sample to explore clinician objections to the test rather than implementation.
2) In the Abstract: Results : 'The implementation study demonstrated reduced rates of antibiotic prescription and follow-up consultation amongst intervention sites.' These results are 'hidden' in Appendix A. I wonder if this should be part of the main body text?	Thank you for this comment. The implementation case study presented was not a direct part of this programme of work and hence is presented in the appendix. We have
	removed the reference to these results from the abstract (lines 49-50)
3) Can the authors say anything about the infection burden during the study period? For example, was there (excess) influenza circulating in the communities sampled? Would the use of CRP be viewed as more valuable during an influenza outbreak and could this possibly affect the views of respondents (in this study)?	The utility of the CRP test would not be limited to influenza outbreaks although consultations with LRTI are likely to be higher at this time. The case study ran from November 2016 – January 2017. This was the shoulder season for influenza that year with levels only exceeding the baseline threshold at the end of December (Source RCGP RSC report).

4) I wonder if the authors could possibly comment on, or have considered, the implementation of influenza and/or respiratory syncytial virus PoC testing instead of CRP PoC testing for LRTI in primary care? Would this be more interesting than CRP?	The focus of the study was on implementation of CRP testing. There is a body of evidence as well as guidelines suggesting CRP testing should be implemented in the NHS. It would be interesting to see the impact of RSV and influenza testing in a primary care setting but this was not the focus of this study.
5) Respondents understand about the association between non-targeted antibiotic prescribing and increased antibiotic resistance, but this may seem as an abstract concept without solid evidence about the (cost) effectiveness of non-targeted antibiotic prescribing within their own practice. Have the authors considered utilising information on the incidence of antibiotic resistance within the local community and then specifically focusing their NPT implementation efforts to regions where the (threat of) antibiotic resistance is highest?	Other studies have utilised practice level prescribing data and feedback to attempt to influence prescribing. Community resistance levels remain low and routine data on resistance is hampered by sampling bias (sampling after response failure), the authors are not aware of studies including local resistance rates as part of the feedback to practitioners.
6) I think that the 'use of electronic alerts during consultation to prompt clinicians to PoC if indicated.' Is an interesting suggestion. However, unless there is a standard consensus on what criteria actually warrant the use of a PoC (not forgetting cost calculations, test specificity for a particular disease, the actual 'weight' of the PoC result in the individual doctor's decision-making process etc), the development of such algorithms may not be as easy to implement as initially appears.	We accept this suggestion may not be successful and that more clinical information would be needed to generate such alerts.
7) The idea of utilizing an 'enthusiastic, local champion to catalyse support for implementation' (in this case for CRP PoC testing) is an interesting idea, but who will the champion represent – Respondents, financial authorities, pharmacists, PoC diagnostic developers or patients?	Indeed. The suggestion arose following the implementation case study in which change was driven by a local champion who sought funding for the consumables and organised
These different potential users may have conflicting views.	locally the implementation of the testing. We suggest this model would need further testing before widespread uptake could be recommended.

8) Within the framework of a 'local champion', it may be worth remembering that local hospitals may represent an untapped pool of knowledge and tips on PoC implementation, having possibly gone through the process of assessing the implementation of PoC themselves (https://www.futuremedicine.com/doi/pdf/10.2217/fmb2016-0120).	Thank you for this suggestion.
9) Reference 29 does not lead to the 'online training course for general practitioners on the use of CRP, a clinical audit form and patient information leaflet' mentioned in the text. This material should be made available for reviewers and readers.	Thank you for this we have now amended the link: https://www.clahrcwessex.nihr.ac.uk/theme/project/20. The resources mentioned in the paper can be found via links in the sidebar to the right of the page, and in the 'More information' section at the bottom of the page.

Reviewer: 2 Reviewer Name: Hasse Melbye Please leave your comments for the authors below This is a study on an attempt to implement CRP test in British general practice. Why the attempt did not succeed was examined by a qualitative study with a strategic sample of interviewees. The study should be of particular interest for GPs in UK, and of primary care policy makers in UK and other countries. For most readers outside Britain the study may appear less relevant. The background for the study is well described, but it would be of help to get more key information on British primary care, for instance how the performance of laboratory tests is organized. In other countries, at least in Scandinavia, an office laboratory is running all the day, and laboratory tests are something GPs refer to, not carry out themselves. The manuscript is generally clearly written, but knowledge on NHS and the funding of primary care in UK would help the reader. Other comments. In the qualitative part of the study, time constraints and facility and placement of testing device were listed among the barriers. In the discussion,	Thank you for these suggestions. In the UK access to health care is free at the point of contact and funded through central taxation. There is no co-payment or health insurance needed for access to the NHS. Practices rarely perform testing on the primary care site with few exceptions (urine dip tests, pregnancy tests and sometimes coagulation monitoring). The majority of lab tests are organised by the local hospital. Patients either have samples taken at the practice and sent to the lab or attend the hospital for sampling. Payment for the tests is by 'block contract' paid for by the commissioning group so neither the patient nor the practice have to bear any cost. If near patient tests are performed like CRP the practice are unable to charge the patients and so
--	--

the funding attained most of the focus. As a reviewer from outside Britain it seems strange that the organization of laboratory services in British primary care escapes attention.	would bear the full cost of the tests from their own income. So near patient testing is unfamiliar to practitioners- hence the need to examine ways to implement the test which will take additional time and cost the practices money. We have added this information in a second appendix at lines 330-341 so that the international reader can access those details to help understand the barriers to implementation. We have also added a reference to the appendix at the relevant point in the main body of text, at line 111.
Main concern. The study carried out in Herts, is presented in Box 2 and the appendix. It is not described in other parts of the manuscript, and its role in a potential paper remains too unclear.	Thank you for this; we have now included reference to the case study in the main body of text, at lines 170-177.
The "no" for statistics in the review check list is due to the missing description of methods for Herts study.	Thank you for this; we have now included a more detailed description of the statistical methods used for the Herts study in the appendix, at lines 347-357.
FORMATTING AMENDMENTS FROM EDITORIAL OFFICE: - Please ensure that your CORRESPONDING AUTHOR in your main document and Scholar One submission system are the same. If more than one author needs to share credit as first or senior author then to have a footnote saying 'xx and yy contributed equally to this paper' instead of listing two corresponding authors. Please refer to below sample: Corresponding author: Author 1 (name and email address) as show in Scholar One. Author 1 and Author 2 contributed equally to this paper. - Please re-upload your supplementary files in PDF format. - We have implemented an additional requirement to all articles to include 'Patient and Public Involvement' statement within the main text of your main document. Authors must include a statement in the methods section of the manuscript under the sub-heading 'Patient and	This point has been addressed. This point has been addressed. Thank you for this; a section on patient and public involvement has been added to the methods, at lines 180-184.

Public Involvement'. This should provide a brief response to the following questions:

How was the development of the research question and outcome measures informed by patients' priorities, experience, and preferences?

How did you involve patients in the design of this study?

Were patients involved in the recruitment to and conduct of the study?

How will the results be disseminated to study participants?

For randomised controlled trials, was the burden of the intervention assessed by patients themselves? Patient advisers should also be thanked in the contributorship statement/acknowledgements. If patients and or public were not involved please state this.

VERSION 2 – REVIEW

REVIEWER	John Hays Erasmus MC, Erasmus University Medical Centre, Rotterdam, the Netherlands
REVIEW RETURNED	17-Jul-2018
GENERAL COMMENTS	This is a re-review of a revised manuscript.